# Current Antimicrobial Use in Horses Undergoing Exploratory Celiotomy: A Survey of Board-Certified Equine Specialists

**DOI:** 10.3390/ani13091433

**Published:** 2023-04-22

**Authors:** Meagan Rockow, Gregg Griffenhagen, Gabriele Landolt, Dean Hendrickson, Lynn Pezzanite

**Affiliations:** Department of Clinical Sciences, College of Veterinary Medicine and Biomedical Sciences, Colorado State University, Fort Collins, CO 80523, USA; meagan.rockow@rams.colostate.edu (M.R.); gregg.griffenhagen@colostate.edu (G.G.); gabriele.landolt@colostate.edu (G.L.); dean.hendrickson@colostate.edu (D.H.)

**Keywords:** horse, antimicrobials, colic, celiotomy, survey

## Abstract

**Simple Summary:**

Recommendations for antimicrobial prophylaxis are well described for abdominal surgery in human medicine, but the information is limited for equine veterinary practice. In addition, recent studies support a reduced duration of antimicrobials postoperatively in horses undergoing celiotomy compared to what has been previously reported; however, protocols vary widely between practices. The overall objective of this study was to provide an updated characterization of the ‘current state of play’ of antimicrobial use in horses undergoing emergency colic surgery and the perceived risk of postoperative complications. Specifically, the aim was to poll veterinary internists and surgeons to determine current usage among respondents in the United States. Improved knowledge of recommendations for antimicrobial prophylaxis for commonly performed procedures in equine practice, such as abdominal surgery, may help to reduce postoperative complications and enhance antimicrobial stewardship at a time when resistance is increasing in equine practice.

**Abstract:**

In the past decade, there has been a considerable increase in the recognition of antimicrobial resistance in equine practice. The objective of this study was to survey the current clinical use of antimicrobials for a commonly performed surgical procedure (exploratory celiotomy) with the goal of understanding how recent literature and changes in microbial resistance patterns may have impacted antimicrobial selection practices. An electronic survey was distributed to veterinary professionals within the American College of Veterinary Internal Medicine (ACVIM) and the American College of Veterinary Surgery (ACVS). A total of 113 completed surveys were returned. Practitioners reported antimicrobials were most frequently given 30–60 min preoperatively (63.1%). Two antimicrobial classes were typically administered (95.5%), with gentamicin (98.2%) and potassium penicillin (74.3%) being the most common. Antimicrobials were typically not re-dosed intraoperatively (78.6%). Factors that affected overall treatment length postoperatively included resection (81.4%), bloodwork (75.2%), enterotomy (74.3%), fever (85.0%), incisional complications (76.1%), and thrombophlebitis (67.3%). The most common duration of antimicrobial use was 1–3 d for non-strangulating lesions (54.4% of cases) and inflammatory conditions such as enteritis or peritonitis (50.4%), and 3–5 d for strangulating lesions (63.7%). Peri-incisional and intra-abdominal antimicrobials were used by 24.8% and 11.5% of respondents, respectively. In summary, antimicrobial usage patterns were highly variable among practitioners and, at times, not concordant with current literature.

## 1. Introduction

Antimicrobial administration in human and veterinary medicine has led to important advancements in patient care. However, inappropriate antimicrobial usage has been shown to contribute to bacterial drug resistance and is not without risk of complications. Guidelines for use aim to maximize therapeutic efficacy while reducing the negative impacts on public health. Perioperative prophylaxis is the most common reason for use of antimicrobials [1,2,3] and has been shown to decrease surgical site infection incidence, postoperative complications, and the cost of treatment [4,5,6]. However, reported compliance with prophylaxis guidelines in veterinary and human practice varies [7,8]. For example, one study indicated that only 6.3% of horses undergoing surgery received preoperative antimicrobials within 60 min of the initial incision [8].

Exploratory celiotomy (‘colic’) surgery in horses is generally considered a ‘clean contaminated’ procedure, although a minority of cases may be considered either ‘clean’ if performed electively and without enterotomy/resection or ‘dirty’ following intra-abdominal contamination or intestinal perforation [9,10,11]. Current recommendations for antimicrobial use in ‘clean-contaminated’ procedures include: (1) perioperative antimicrobial prophylaxis is indicated; (2) broad-spectrum selection is recommended but should consist of lower-generation drugs to minimize the emergence of resistant bacterial strains; (3) selection should be based on commonly identified isolates; (4) the intravenous route of administration within 30 to 60 min prior to incision is recommended; (5) antimicrobials should be re-dosed intraoperatively if surgery continues beyond two half-lives; and (6) aseptic technique should be followed. Complications after colic surgery that may be reduced with appropriate antimicrobial protocols reportedly include surgical site infection, thrombophlebitis, pneumonia, peritonitis, colitis, and *Salmonella* shedding [9,12,13,14].

The most recent available review of antimicrobial prophylaxis for exploratory celiotomy in horses at one large university referral practice was reported in 2012 [9] and concluded that the majority of horses treated for surgical colic at that facility received inaccurate antimicrobial prophylaxis, both in terms of the dose received and the timing of drug administration [9]. More recent studies have demonstrated no added beneficial effect of 120 h vs. 72 h or a single preoperative dose of antimicrobials to reduce the incidence of surgical site infection [11,15]. Many institutions base antimicrobial usage guidelines on recommendations from human practice, the impression of common practice within veterinary medicine, or consensus statements when available, as clinical trials in equine practice comparing antimicrobial protocols for gastrointestinal surgery are limited [8,16,17,18,19].

Therefore, the overall aim of this study was to provide an updated examination of antimicrobial usage for abdominal surgery in horses. Specifically, the aim was to poll members of the American College of Veterinary Internal Medicine (ACVIM) and American College of Veterinary Surgeons (ACVS) to determine current antimicrobial usage in the United States as well as their perceived frequency of postoperative complications. We hypothesized that despite published recommendations for clean-contaminated procedures such as exploratory celiotomy in horses, antimicrobial practices would vary between clinicians.

## 2. Materials and Methods

### 2.1. Prospective Polling of ACVIM and ACVS Diplomates on Current Antimicrobial Practices

A survey was posted on the ACVIM listserve, and the ACVS college was polled via email to survey specialists on their current antimicrobial administration practices prior to and following exploratory celiotomy in horses. Information requested included preoperative drug/dose information, number of antimicrobial classes administered, time from preoperative dose to surgical incision (recorded or intended), whether antimicrobials are routinely re-dosed during surgery, and if so at what time point following the first dose, duration of use, and whether postoperative use is dictated by factors specific to an individual case such as procedure performed (e.g., enterotomy, bowel resection) or bloodwork (e.g., complete blood count indicating normal leukocyte and neutrophil counts), whether peri-incisional or intra-abdominal antimicrobials were administered, and incidence of postoperative complications (e.g., fever (>101.5 °F), incisional complications, thrombophlebitis, or other catheter associated complications and *Salmonella* shedding) is known for that practice.

### 2.2. Data Analysis

The respondents were able to submit an incomplete survey if they so desired. When appropriate, the respondents could provide more than one response to a specific question. Pearson’s chi-squared test and Fisher’s exact test were performed to investigate the association between predictors (years in practice, ACVIM vs. ACVS college) and other reported binary outcomes (intraoperative redosing, peri-incisional, or intra-abdominal antibiotics). Prism software v8.4.1 (GraphPad Software Inc., La Jolla, CA, USA) and R version 4.1.2 (R Foundation for Statistical Computing, Vienna, Austria) were used for graph generation and statistical analyses, with significance assessed as *p* < 0.05.

## 3. Results

### 3.1. Demographic Information

A total of 113 completed surveys were returned (68.8% ACVS/ECVS [n = 1 ECVS with the remainder being ACVS] and 31.2% ACVIM/ECVIM [n = 1 ECVIM with the remainder being ACVIM]). As the surveys were distributed both to a listserve and an email distribution list, the total number of surveys sent and therefore the total response rate were unable to be determined. The number of years in practice were relatively evenly distributed between experience group levels, with 38.6% of respondents having >20 years experience, 18.4% having 15–20 years, 20.2% having 10–15 years, 18.4% with 5–10 years, and 4.4% with 0–5 years (Figure 1). However, although the group with the fewest years of practice was subjectively underrepresented, no associations were detected between diplomate college or years of practice and the variables examined (Table 1).

### 3.2. Antimicrobial Selection and Timing

The practitioners reported that preoperative antimicrobials were most commonly administered 30 to 60 min prior to first incision (63.1%), with <30 min in 28.8% of cases and >60 min in 2.7%, and were not recorded or varied between cases for 5.4% of respondents. Two classes of antimicrobials were given in most cases (95.5%), with gentamicin (6.6–8.8 mg/kg) and potassium penicillin (22,000–44,000 IU/kg) being administered most frequently (98.2% and 74.3%, respectively). Other commonly reported antimicrobials administered included procaine penicillin G (25,000 IU/kg) (31.9%), ceftiofur (2.2–5.0 mg/kg) (19.5%), and cefazolin (11.0–22.0 mg/kg) (8.0%) (Figure 2).

### 3.3. Antimicrobial Dosing

Antimicrobial doses were based on preoperative weight, determined by the scale in all horses in 26.5% of cases or, in some cases, pending patient comfort level in 53.1% of cases. In some situations, the weights were determined by weight tape (9.7%) or a visual estimate (10.6%). Antimicrobials were most commonly not re-dosed intraoperatively (78.6% of cases). In those instances where antimicrobials were re-dosed, the timing of the second dose was determined by 2 times x the half-life of the antimicrobial (50%), based on the normal drug dosing interval (33.3%), or based on a different drug dosing interval (16.7%) (Figure 3).

### 3.4. Treatment Length of Antimicrobial Administration

The preoperative or intraoperative factors that affected the duration of antimicrobial administration were reported to include bowel resection (81.4%), bloodwork findings (75.2%), and enterotomy (74.3%). Postoperative factors that affected the duration of antimicrobial administration included fever (85.0%), incisional complications (76.1%), catheter-associated complications, including thrombophlebitis (67.3%), bloodwork findings (46.0%), colitis (etiology other than *Salmonella*, 32.7%), and *Salmonella* shedding (15.0%). The respondents also self-reported that clinical impression (2.7%), documented evidence of pneumonia (6.2%), or peritonitis (8.0%), in addition to fever, were aspects of case management that may affect the duration of antimicrobial administration. The most frequently reported average duration of antimicrobial administration was estimated to be 1 to 3 days for non-strangulating lesions (54.5% of respondents), 3 to 5 days for strangulating lesions (63.7%), and 1 to 3 days for inflammatory disease (50.4%) (Figure 4).

### 3.5. Complications

Self-reported estimated incidence of complications over all surgeries evaluated included fever, incisional infection, incisional herniation/failure, thrombophlebitis, *Salmonella* shedding, and colitis (etiology other than *Salmonella*) (Figure 5).

### 3.6. Peri-Incisional Antimicrobial Administration

Peri-incisional antimicrobials were reportedly used by 24.8% of respondents. Of those who used antimicrobials by this route in their practice, antimicrobials were most frequently implemented greater than 40% of the time (68.8% of respondents). Selection of antimicrobials for the peri-incisional route was highly variable, with amikacin (32.1%), gentamicin (14.3%), and penicillin (14.3%) being most commonly reported (Figure 6).

### 3.7. Intra-Abdominal Antimicrobial Administration

Intra-abdominal antimicrobials were reportedly used by 11.5% of respondents. Of those who used antimicrobials by this route, the frequency of use was variable, with 35.3% using antimicrobials in <10% of cases, 10–20% (11.8%), 20–40% (17.6%), and >40% of cases (35.3%). The most commonly used drugs included penicillin (58.3%) and gentamicin (16.7%) (Figure 7).

## 4. Discussion

This study contributes to the current understanding of antimicrobial use in the perioperative period for exploratory celiotomy by equine surgeons and internal medicine specialists. As antimicrobial resistance is considered an emerging ‘One Health’ medical issue in both human and veterinary medicine, periodic auditing of clinical practices is warranted. The results of this survey highlight significant variation in approaches to antimicrobial administration, similar to previous studies detailing differences between institutions as well as between intended and actual use [9,16,20,21]. These findings represent a description of current practice rather than evidence to support antimicrobial dosing recommendations; however, surveys of clinical practice have been used as an established approach in human medicine to guide informed decision-making when there is insufficient data from case-controlled clinical trials to provide evidence-based guidelines, as is the case for many instances where antimicrobials are used in equine clinical practice [22]. The limited number of antimicrobials approved for use in horses presents a challenge to equine practitioners, resulting in frequent extra-label and compounded drug use [23]. Although survey enrollment in this study was limited, as has been the case with other recently published analyses of equine practice [24,25,26], these findings represent an updated picture of the clinical use of antimicrobials by equine specialists in the United States.

Guidelines for antibiotic selection in equine patients have been adapted from human medicine and typically include the implementation of early-generation broad-spectrum antimicrobials given intravenously prior to surgery [1,5,20,27]. The findings of this study were generally in concordance with that recommendation as well as previous reports [9], with two drug classes being administered most commonly preoperatively, frequently potassium penicillin and gentamicin [28]. As surgical site infections (SSI) are the second most commonly reported short-term complication following celiotomy (after persistent postoperative signs of colic) [29], antimicrobial selection should be dictated by efficacy against common pathogens. Previous reports of SSI in horses have identified *Enterobacteriaceae*, *Enterococcus*, *Staphylococcus, and Streptococcus* as commonly identified bacterial isolates [30,31,32].

Aminoglycosides such as gentamicin are effective against *Enterobacteriaceae* and, in some instances, *Staphylococci*, although systemic administration is considered extra-label use as the only FDA-approved indication for horses is via intra-uterine infusion [28]. Penicillin is also used extra-label to treat infections in horses, including *Streptococcus*. However, in one study evaluating all isolates cultured in equine celiotomy SSI, penicillin-resistant isolates accounted for 92%, while an additional 18% were gentamicin-resistant [30,31,32]. Other studies have corroborated these findings, indicating a significant increase in the percentage of resistant equine isolates over time [33,34,35,36,37] and greater consideration of drugs considered to be critically important for human use [1]. Additionally, positive intraoperative cultures of the incision have not been shown to be predictive of SSI, and when SSI did occur, it was due to a different bacterial isolate in that study [30,31]. Furthermore, a variety of bacterial species may be isolated from laparotomy incisions peri-operatively without the development of SSI, indicating that while contamination of the incision peri-operatively may be one mechanism by which SSI occurs, evaluation of other mechanisms such as bacteremia postop warrant further investigation [30,31]. While the selection of early-generation broad-spectrum antimicrobials appears appropriate based on a recent review of the literature for ‘clean-contaminated’ procedures, further consideration of the timing of the first administration and postoperative duration is warranted.

The timing of initial antimicrobial administration has been recommended to be within 60 min before the first incision based on human guidelines [37] but has historically been cited as challenging in equine practice for a variety of reasons, including hospital policies, the emergent nature of surgery in some instances, and concern for hypotension when administered following anesthetic induction [9,20,21]. In one tertiary equine referral hospital where clinical audits of antimicrobial recording were performed to raise awareness of the timing of antimicrobial dose prior to surgery, improvement was shown between audits for elective arthroscopies but not for emergency laparotomies, highlighting that in many cases, lack of compliance with drug guidelines is likely the result of challenges faced by personnel working in critical conditions rather than a lack of awareness or an unwillingness to comply [20]. Evidence in humans undergoing emergency surgery supports the concept that achieving high concentrations of antimicrobials at the time of incision and throughout surgery is likely more important in reducing infection than the duration of antimicrobial therapy postoperatively [38], which encourages increased awareness of drug timing relative to surgery when possible.

Intraoperative redosing of antimicrobials for prolonged surgical procedures where concentrations of prophylactic antimicrobials may decrease below MIC for common pathogens has been described as similarly crucial to initial drug administration in human surgery to reduce infection [39]. Human guidelines indicate redosing is warranted for procedures lasting longer than two drug half-lives or with significant blood loss (i.e., in humans >1.5 L) [40,41,42]. Further evidence to support this concept has been demonstrated in multiple reports [43,44,45] and ongoing clinical trials assessing optimal timing for abdominal surgery specifically [46]. However, noncompliance with this aspect of antimicrobial use has been similarly recognized in multiple studies as an area where improvement may be made [47,48,49,50]. Similarly, intraoperative redosing is reported to be infrequently performed in equine colic surgery [9,19], including in the authors’ practice, which was also reflected in the results of this survey. The half-life of penicillin, one of the most commonly used drugs in this survey and a time-dependent antimicrobial, is 30 to 40 min in horses, indicating subtherapeutic levels may be reached during critical stages of longer procedures, such as resection and anastomoses, and recovery from general anesthesia [9,19]. The rationale for not performing intraoperative redosing has been cited as being due to the perceived risk of hypotension, particularly with penicillin drugs, by anesthesiologists, a lack of awareness of drug half-lives by clinicians, or inadvertent noncompliance intraoperatively [9,51]. However, evidence in the human literature suggests that stricter adherence to appropriate intraoperative redosing recommendations may reduce the risk of incisional infection, although this requires further investigation in equine practice specifically [19]. Compliance with redosing of intraoperative antimicrobials in human surgery has been shown to be objectively improved through a combined approach involving the implementation of clearly defined guidelines, increased education of healthcare providers, and automated paging systems prompting redosing, which could also be integrated into veterinary surgery [52,53,54].

General guidelines regarding antimicrobial duration in veterinary medicine support the idea that they should be administered for the shortest effective duration to reduce the risk of the development of resistant pathogens [27,55]. In equine celiotomy procedures, duration of administration has anecdotally been administered to reduce the risk of surgical site infection, pneumonia, and peritonitis. When considering surgical site infection specifically, prolonged antimicrobial use beyond 24 h postoperatively in humans did not reduce the incisional complication rate even in surgeries classified as ‘dirty’ [40]. This concept has been supported by several recent papers in the equine literature that did not demonstrate additional benefits to prolonged antimicrobial administration following equine laparotomy [11,15]. Durward-Akhurst reported comparable incisional complication rates with 72 versus 120 h of antimicrobial administration, concluding that no benefit of using a prolonged period of antimicrobials should be expected [15]. As colic surgery is generally considered a clean-contaminated procedure, protocols in human medicine would suggest that it is possible to reduce use to 24 h postoperatively [5]. No additional benefit beyond a single prophylactic dose versus 120 h of administration was found in another recent pilot study in horses, although it was acknowledged that the power calculation indicated that a larger sample size enrolled over a longer study period would be necessary to draw general conclusions [11]. Another study evaluating temporal changes in antimicrobial regimens for equine colic surgery also indicated no additional benefit to administering antimicrobials past a single preoperative dose [55]. However, whether these findings can be extrapolated to equine patients undergoing enterotomy or resection and anastomosis without increased risk for peritonitis requires further investigation. Additionally, as intubation may be performed in painful patients without standard oral rinsing in some cases, increased gross contamination of the lower airway may prompt longer antibiotic durations postoperatively to reduce the perceived risk of pneumonia. Furthermore, individual case factors (e.g., pyrexia, incisional or catheter-associated complications, bloodwork, preoperative immune statuses such as pars pituitary intermedia dysfunction, or perceived risk of pulmonary complications from intubation) are frequently considered by clinicians when determining antimicrobial duration, as was reflected in this report, rather than applying standard lengths of time for each case. Finally, it is recognized that other factors related to case management are within the clinician’s control (e.g., subcutaneous lavage, skin closure techniques, abdominal bandaging). However, antimicrobial duration also plays a role in reducing surgical site infection [56].

Another consideration that has been raised with the extended duration of antimicrobial administration is regarding the prevention of intra-abdominal adhesion development, as one study indicated that administration of the combination of 72 h of potassium penicillin, gentamicin, and flunixin meglumine prevented abdominal adhesion formation in foals with experimental ischemia [57]. However, a shorter duration of antimicrobial administration was not evaluated, and therefore conclusions could not be definitively drawn regarding the optimal drug administration length to reduce adhesions [57]. Taken together, these findings support the notion that, as most colic surgeries are classified as ‘clean-contaminated,’ antimicrobial prophylaxis is indicated, but that extended duration may not be necessary, and larger studies are needed to draw conclusions regarding protocols for specific indications. Given the expense of prolonged antimicrobial treatment and increased concern with antimicrobial resistance, a more selective approach to drug administration is indicated [19].

Postoperative complications most frequently reported in this survey included fever and incisional infection. Of note, the prevalence of mild postoperative pyrexia (>38.3 °C) is reportedly high (85%) in the early postoperative period (<48 h) and does not equate to bacterial infection nor the necessity for long-term antimicrobial use necessarily [5,19]. Prevalence of SSI has been previously reported to be 10–42%, similar to findings self-reported here as predominantly <10 or 10–20% of cases, although variability in reporting is likely due in part to different definitions of SSI used, the fact that estimates were reported here rather than recorded incidences, and follow-up available to clinicians upon dismissal from the hospital [15,19,29,30,31,58]. Infection is known to impact the length of stay and therefore the cost of care and result in reduced long-term survival [19]. Notably, client-reported satisfaction with the celiotomy procedure was not shown to be reduced with the development of postoperative infection, except in cases of *Salmonellosis* in one study at a tertiary referral hospital [19].

The incidence of postoperative complications, in addition to survival rate and prognostic indices (which were not surveyed in this study), may not be comparable across regions due to differences in management, climate, or diet (e.g., types of forage) that may influence the onset of abdominal colic [59,60,61,62]. However, in this study, several common complications were reported with a similar frequency to recent reports [63]. Other complications that have been infrequently reported following colic surgery were not specifically questioned in this survey nor mentioned by participants, including penicillin-induced immune-mediated hemolytic anemia [11,64], sinus infection [65], hemoperitoneum [63], or those that may be perceived as secondary to the anesthetic event itself (e.g., myopathy, fracture). Furthermore, it is acknowledged that complication rates (e.g., for surgical site infection) postoperatively have been shown to be influenced by multiple other variables outside of the clinician’s control, which could not be accounted for in this survey format. These include such factors as the season of the year, body weight, age, duration of colic signs, physical examination findings, bloodwork, peritoneal fluid analysis findings on admission, whether small intestinal resection was necessary, or whether the emergent nature of the condition necessitated surgery outside of normal working hours [30,31,56,65,66,67,68]. Due to the factors discussed, reported complication rates were not correlated with individual antimicrobial protocols in this survey.

Additional antimicrobial therapy via routes other than systemic administration (i.e., peri-incisional and intra-abdominal) was reported in 25 and 12% of cases in this survey, respectively, although survey respondents were not asked to clarify whether the usage was intended to prevent or treat infection (i.e., intra-abdominal use in cases where celiotomy was performed to treat established peritonitis) [69]. Evidence for administration by these routes is limited in equine practice, and appropriate doses to minimize local cytotoxicity have not been determined given that administration by either route is extra-label. Antimicrobial inclusion in lavage fluids in experimental studies in horses was reported to induce a mild, transient inflammatory response to peritoneal fluid, although efficacy has not been definitively assessed in equine studies to the authors’ knowledge [70]. Povidine-iodine solutions administered intraabdominally further induced chemical peritonitis in horses [70]. Recent meta-analyses in human surgery may support a potential benefit to intra-operative intra-cavitary lavage and wound/incision irrigation, with reduced SSI in groups receiving any surgical site lavage versus none, and those receiving antibacterial versus non-antibacterial lavage interventions, antibacterial versus povidine-iodine or saline, and pulsatile versus standard lavage, although reports included were conflicting [71,72]. However, current guidelines by the Surgical Infection Society on the management of abdominal infection do not include recommendations for antimicrobial use beyond systemic administration [73]. Further investigation of adjunctive antimicrobial techniques in equine practice for abdominal surgery would be warranted in light of antimicrobial stewardship and to assess local cytotoxicity and efficacy prior to administration.

The limitations of this study include the self-reported nature of the findings and the likely low survey response rate. Whether individual clinicians based their responses on objective personal audits performed recently or were relying on clinical impressions of their practice was not recorded. The number of survey responses received was lower than expected, indicating that a minority of the populations contacted responded, which is similar to other recently published surveys in equine practice [24,25,26]. In addition, the electronic survey was sent only to ACVS and ACVIM diplomates, which may have been selected for response bias, and it is acknowledged that the results may not reflect usage by individuals in other colleges or non-specialists who may use antimicrobials perioperatively for equine celiotomy. When assessing complications, other factors that may influence the surgical site infection rate, for instance, were not surveyed as antimicrobial usage was the primary focus of this work (e.g., abdominal bandaging, skin closure technique, and suture selection, quality of anesthetic recovery, IV fluid therapy, intraoperative arterial pressure) [19,57,74,75,76,77]. Not all participants answered all questions posed, indicating some participation bias. Furthermore, the study, as designed, likely did not capture all possible complications or those infrequently reported (e.g., sinus infection, pneumonia, hemoperitoneum) and did not consider potential complications of any surgical procedure under general anesthesia in horses (e.g., myopathy) [63,65]. Despite these limitations, this study provided important insight into the current clinical use of antimicrobials for equine abdominal surgery by large animal specialists.

## 5. Conclusions

Clinical auditing has been demonstrated to improve the quality of patient care in multiple care settings [78,79] and may be applied in equine celiotomies to improve the completeness of record keeping and raise awareness of current recommendations for antimicrobial prophylaxis. Identifying members of the anesthesia and/or perioperative care teams as responsible for antimicrobial administration may bring the timing of the first dose closer to the surgical incision. A greater emphasis on the timing of surgical preparation (e.g., removing hair or surface contamination overlying the celiotomy incision site) prior to induction and premedication may reduce the lag time between antimicrobial administration and the initiation of surgery. However, it is recognized that the lack of improvement in one study with antimicrobial administration for emergency procedures following a clinical audit [20] highlights the complexity of the situation for clinicians treating emergency celiotomy. The consciousness of antimicrobial protocols in light of increased antimicrobial resistance in veterinary practice and the completeness of medical records from a legal perspective is of mounting importance [20]. With this in mind, periodic audits of actual versus intended practices may provide clinical benefits to improve fiscal responsibility to clients, reduce antimicrobial-associated complications, and improve antimicrobial stewardship in light of increased drug resistance. Larger case-controlled clinical trials in horses are indicated to determine and compare the efficacy between antimicrobial protocols with regard to intraoperative redosing and antimicrobial duration postoperatively for specific surgical indications.

## Figures and Tables

**Figure 1 animals-13-01433-f001:**
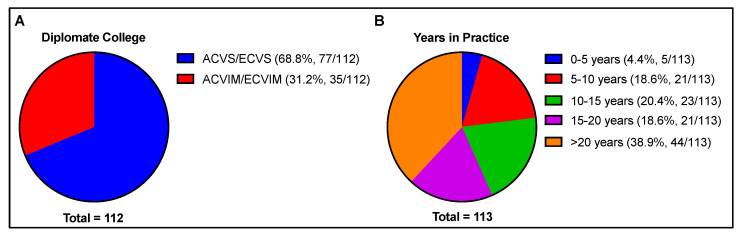
Demographic information of respondents, including (**A**) diplomate college and (**B**) years in practice. Total number of responses per question indicated beneath each chart.

**Figure 2 animals-13-01433-f002:**
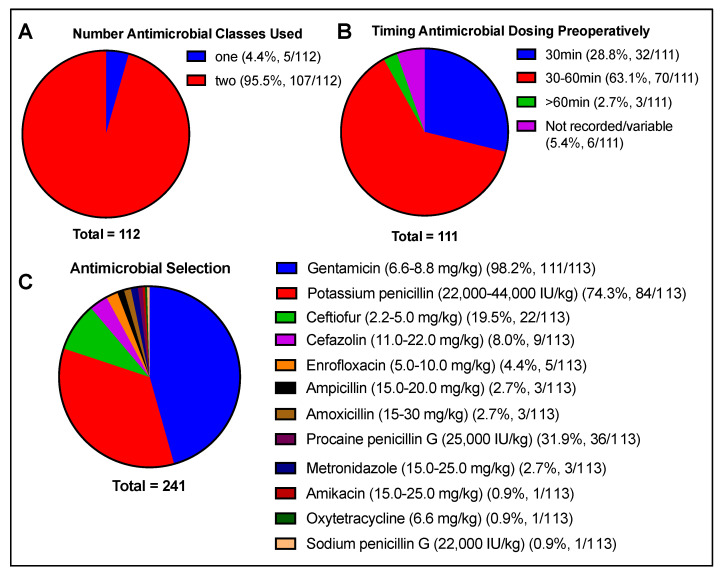
Antimicrobial selection and timing of dosing preoperatively, including (**A**) number of antimicrobial classes used, (**B**) timing of antimicrobial dosing preoperatively, and (**C**) antimicrobial selection. Total number of responses per question indicated beneath each chart. Respondents were not limited in the number of responses to antimicrobial selection.

**Figure 3 animals-13-01433-f003:**
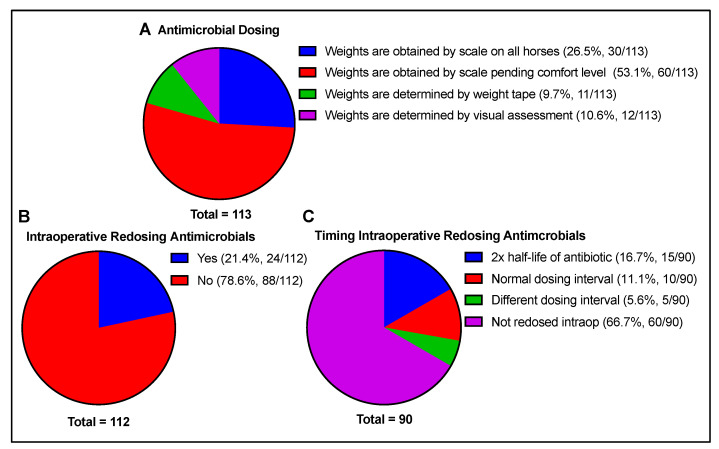
Antimicrobial dosing details include (**A**) how doses are determined preoperatively, (**B**) whether intraoperative redosing is performed, and (**C**) the timing of intraoperative redosing if performed. Total number of responses per question indicated beneath each chart.

**Figure 4 animals-13-01433-f004:**
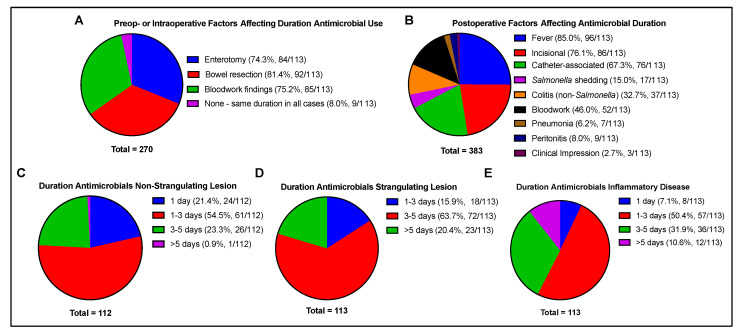
Factors affecting the duration of antimicrobial use, including (**A**) preoperative or intraoperative factors, (**B**) postoperative factors, and type of primary lesion determined intraoperatively, including (**C**) non-strangulating lesions, (**D**) strangulating lesions, and (**E**) inflammatory disease. Total number of responses per question indicated beneath each chart. For (**A**, **B**), percentages are indicated out of the total number of individuals responding (n = 113) rather than total responses.

**Figure 5 animals-13-01433-f005:**
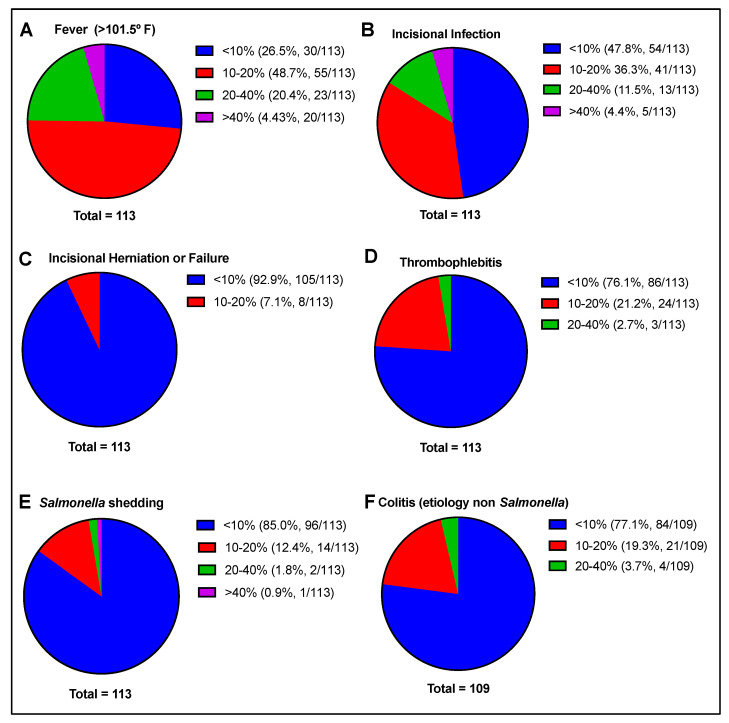
Reported rate of complications postoperatively, including (**A**) fever, (**B**) incisional infection, (**C**) incisional herniation or failure, (**D**) thrombophlebitis, (**E**) *Salmonella* shedding, and (**F**) colitis (other than *Salmonella*). Total number of responses per question is indicated beneath each chart.

**Figure 6 animals-13-01433-f006:**
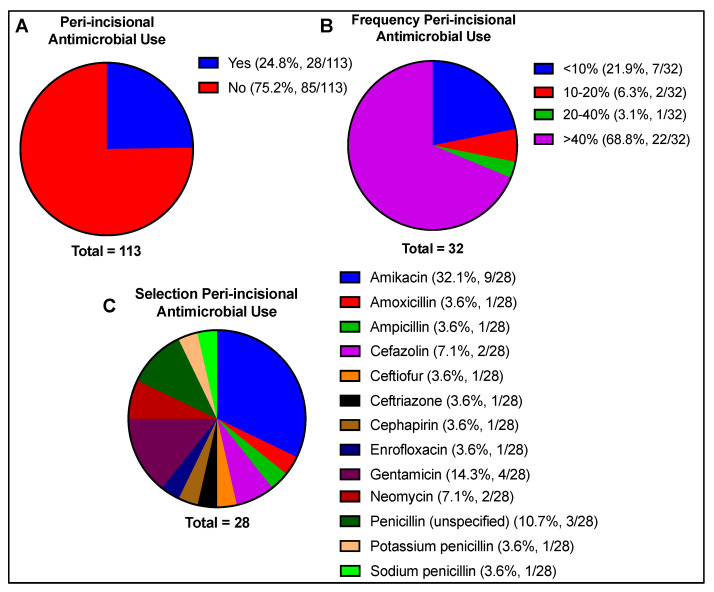
Peri-incisional antimicrobial use, including (**A**) reported usage (yes/no), (**B**) frequency of use if reported, and (**C**) antimicrobial selection if used. Total number of responses per question is indicated beneath each chart. Respondents were not limited in the number of responses to antimicrobial selection.

**Figure 7 animals-13-01433-f007:**
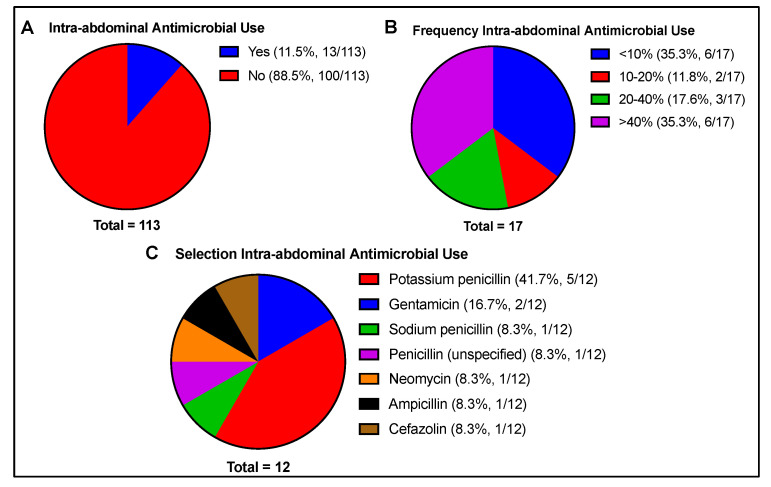
Intra-abdominal antimicrobial use, including (**A**) reported usage (yes/no), (**B**) frequency of use if reported, and (**C**) antimicrobial selection if used. Total number of responses per question is indicated beneath each chart. Respondents were not limited in number to antimicrobial selection.

**Table 1 animals-13-01433-t001:** Comparison of survey responses separated by diplomate college (ACVIM vs. ACVS) and years in practice for the respondent’s answers to the questions of whether they re-dosed antibiotics intraoperatively, administered antibiotics peri-incisionally, or administered antibiotics intra-abdominally. Outcomes are presented as an absolute number of responses as well as percentages of respondents answering yes or no within each grouping. No significant differences were noted between groups using a *p*-value cutoff of 0.05, and chi-squared *p*-values are presented (Fisher’s exact test *p*-values were also agreed upon but are not shown here).

	Antibiotics Redosed Intraoperatively		Peri-Incisional Antibiotics Administered		Intra-Abdominal Antibiotics Administered	
	No	Yes	No	Yes	No	Yes
Diplomate College						
ACVIM	24 (69%)	11 (31%)	29 (83%)	6 (17%)	31 (89%)	4 (11%)
ACVS	62 (84%)	12 (16%)	53 (72%)	21 (28%)	66 (89%)	8 (11%)
	Chi-squared *p*-value	0.079		0.236		1
Years in Practice						
0–5	4 (80%)	1 (20%)	4 (80%)	1 (20%)	5 (100%)	0
5–10	14 (70%)	6 (30%)	18 (90%)	2 (10%)	18 (90%)	2 (10%)
10–15	18 (82%)	4 (18%)	15 (68%)	7 (32%)	20 (91%)	2 (9%)
15–20	15 (71%)	6 (29%)	15 (71%)	6 (29%)	19 (90%)	2 (10%)
>20	35 (85%)	6 (15%)	30 (73%)	11 (27%)	36 (86%)	6 (14%)
	Chi-squared *p*-value	0.574		0.525		0.888

## Data Availability

The data is contained within the article.

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
