# Peer review of "Current Antimicrobial Use in Horses Undergoing Exploratory Celiotomy: A Survey of Board-Certified Equine Specialists"

_animals, 2023, doi:10.3390/ani13091433_

Round 1
Reviewer 1 Report
Dear Authors,
Thank you for the opportunity to review this interesting manuscript. I find it informative and well written. As the authors themselves acknowledge, due to the nature of the study and its design, little conclusions can be taken in term of efficacy of antibiotic treatments considered, and their association with reported complications type and rates. Nevertheless, the gathered information is valuable and mostly well presented.
Just a few remarks:
Lines 120-121: I don’t understand this sentence. Do the authors mean that no association was detected between diplomate college or years in practice and the variables examined (antibiotic regimen choice and complications rates)? I would state more clearly which associations were investigated.
Line 157: Salmonellosa shedding. I suggest to rephrase to “Salmonella shedding”. Same at line 175, in Figure 5, and in figure 179, as Salmonellosis is mostly understood as the clinical disease caused by Salmonella
Line 161: was estimated
Figure 4: Spell check peritonitis. What do the authors mean by strangulating displacement? Just strangulating large colon displacements or strangulating lesions of any intestinal segment? If the latter, I would suggest using the broader terms “strangulating lesions” and “non-strangulating lesions”. I also don’t understand the difference between non-strangulating displacement and simple obstruction.
Line 439: spell and punctuation check reference number 8.
Also check the format of the references and make sure it is consistent with the Journal’s guidelines.
Author Response
Thank you to the reviewers for their time and efforts. The authors have responded to each point below individually.
Reviewer 1
Thank you for the opportunity to review this interesting manuscript. I find it informative and well written. As the authors themselves acknowledge, due to the nature of the study and its design, little conclusions can be taken in term of efficacy of antibiotic treatments considered, and their association with reported complications type and rates. Nevertheless, the gathered information is valuable and mostly well presented.
Just a few remarks:
Lines 120-121: I don’t understand this sentence. Do the authors mean that no association was detected between diplomate college or years in practice and the variables examined (antibiotic regimen choice and complications rates)? I would state more clearly which associations were investigated.
- Thank you requesting this clarification – the wording has been changed to reflect the authors’ intent and the reviewer’s comment, ‘No associations were detected between diplomate college or years in practice and the variables examined.’ A table has also been added to illustrate statistical analyses for the reader to interpret.
Line 157: Salmonellosa shedding. I suggest to rephrase to “Salmonella shedding”. Same at line 175, in Figure 5, and in figure 179, as Salmonellosis is mostly understood as the clinical disease caused by Salmonella
- Thank you for this suggestion. This wording has been changed throughout and in Figure 5.
Line 161: was estimated
- This change has been made.
Figure 4: Spell check peritonitis. What do the authors mean by strangulating displacement? Just strangulating large colon displacements or strangulating lesions of any intestinal segment? If the latter, I would suggest using the broader terms “strangulating lesions” and “non-strangulating lesions”. I also don’t understand the difference between non-strangulating displacement and simple obstruction.
- Thank you for requesting this clarification. Spelling of peritonitis has been corrected. Presentation of findings in Figure 4 have been simplified as non-strangulating lesions, strangulating lesions, and inflammatory disease (combining non-strangulating displacements and simple obstructions as reported antimicrobial usage was identical for these groups and they are both non-strangulating as the reviewer has pointed out).
Line 439: spell and punctuation check reference number 8.
- Thank you for catching these errors. This reference has been corrected.
Also check the format of the references and make sure it is consistent with the Journal’s guidelines.
- Reference formatting has been changed throughout to be consistent with journal guidelines.
Reviewer 2 Report
While surveys in general are poor indicators of what is actually done or seen, due to lack of response and poor memory/accuracy, an antibiotic survey has not been published recently. The authors do not overstate the results, and fairly judge their findings in the discussion. While of low benefit to the general body of knowledge, it provides some information to stimulate discussion.
Line 11: can this be clarified “below” to maybe a reduced duration? I doubt you mean below the recommended dosage.
Line 32: can strangulating SI lesions also be included here, as they are fairly common reasons for surgery. Later on, you mention strangulations as a broad group, not just large intestinal. (line 163) Please be sure these are the same.
Line 41: It might be good to divide up this sentence. A more direct sentence structure would improve the manuscript.
Line 62: the comment “in fact” seems to contradict the point of this sentence-antimicrobials do not prevent or treat enteric salmonella, and can contribute to its overgrowth by inhibiting normal flora. Rewording would clarify this sentence better.
Line 72: this sentence is a run on, and makes it difficult to understand what increases or decreases risk. Please reword.
Line 77: does this mean specifically for GI surgery? Clinical evaluation of antibiotic use for elective surgeries have been reported.
Figure 6C is very difficult to read. Is there a better way to present this? The colors are too close together.
Line 175: can this section be better described-are these complications of all surgeries and an estimation of incidence? Did they use charts to identify the complications or memory?
Line 238: this sentence combines two ideas-can this be separated out? Critical use is not related to resistance patterns as discussed here.
Line 293: can the risk of peritonitis and pneumonia also be discussed? SSI is not the only post op complication abx are administered to prevent.
Line 303: is this in horses? Please clarify throughout, as the discussion switches back and forth with humans literature.
Line 367: please reword this sentence. It is unclear whether the addition of abx to the solution makes a difference-it may also be dangerous to suggest this, considering it is off label use and directly counters good antimicrobial stewardship. Please perform a more thorough literature search to better describe the benefits and risks of lavage with antibiotics and or iodine.
Many of the sentences are quite long or run ons. Please revise to be more direct.
Author Response
Thank you to the reviewers for their time and efforts. The authors have responded to each point below individually
Reviewer 2
While surveys in general are poor indicators of what is actually done or seen, due to lack of response and poor memory/accuracy, an antibiotic survey has not been published recently. The authors do not overstate the results, and fairly judge their findings in the discussion. While of low benefit to the general body of knowledge, it provides some information to stimulate discussion.
Line 11: can this be clarified “below” to maybe a reduced duration? I doubt you mean below the recommended dosage.
- Thank you for this suggestion – this has been clarified to reflect reduced duration compared to what has been previously reported.
Line 32: can strangulating SI lesions also be included here, as they are fairly common reasons for surgery. Later on, you mention strangulations as a broad group, not just large intestinal. (line 163) Please be sure these are the same.
- Thank you for this comment. To address this and reviewer 1’s comments above, terminology here and throughout has been clarified as strangulating and non-strangulating lesions, including large and small intestine.
Line 41: It might be good to divide up this sentence. A more direct sentence structure would improve the manuscript.
- This change has been made.
Line 62: the comment “in fact” seems to contradict the point of this sentence-antimicrobials do not prevent or treat enteric salmonella, and can contribute to its overgrowth by inhibiting normal flora. Rewording would clarify this sentence better.
- This has been revised.
Line 72: this sentence is a run on, and makes it difficult to understand what increases or decreases risk. Please reword.
- This change has been made.
Line 77: does this mean specifically for GI surgery? Clinical evaluation of antibiotic use for elective surgeries have been reported.
- This has been clarified to reflect in equine practice for gastrointestinal surgery.
Figure 6C is very difficult to read. Is there a better way to present this? The colors are too close together.
- Thank you for this feedback. In the interest of presenting the findings in a consistent visual manner, the authors would suggest to maintain the current format for 6C (and 2C which may also be viewed as having small numbers and multiple colors). However, the authors will defer to the final editorial opinion as to whether to present the material in its current format or alternatively as a table with the figure.
Line 175: can this section be better described-are these complications of all surgeries and an estimation of incidence? Did they use charts to identify the complications or memory?
- Thank you for requesting this clarification. This section represents a self-reported estimate of incidence of complications of all surgeries by memory. This has been clarified.
Line 238: this sentence combines two ideas-can this be separated out? Critical use is not related to resistance patterns as discussed here.
- This section has been revised.
Line 293: can the risk of peritonitis and pneumonia also be discussed? SSI is not the only post op complication abx are administered to prevent.
- The reviewer raises excellent points here which stimulated further discussion amongst our group as well. In regards to peritonitis, this was the authors’ intent with discussing patients undergoing enterotomy or anastomosis, which has been more clearly stated. Furthermore, clinical scenarios where standard oral rinsing is not performed prior to intubation in painful patients may result in increased lower airway contamination and perceived risk for development of pneumonia which may prolong antibiotic therapy duration. This section has been revised to reflect peritonitis and pneumonia as other postoperative complications for which antibiotics are administered to prevent and warrant further investigation, despite literature to date focusing heavily on surgical site infection as a primary measurable outcome.
Line 303: is this in horses? Please clarify throughout, as the discussion switches back and forth with humans literature.
- This has been clarified to reflect in horses.
Line 367: please reword this sentence. It is unclear whether the addition of abx to the solution makes a difference-it may also be dangerous to suggest this, considering it is off label use and directly counters good antimicrobial stewardship. Please perform a more thorough literature search to better describe the benefits and risks of lavage with antibiotics and or iodine.
- Thank you for this suggestion and requesting this clarification. It was not the authors’ intention to recommend use by other routes; additional references and clarification has been added to improve this section.